# Seasonal Variability of Retroflection Structures and Transports in the Atlantic Ocean as Inferred from Satellite-Derived Salinity Maps

**Paola Castellanos** [1,2,*], **Estrella Olmedo** [1,3], **Josep Lluis Pelegrí** [1,4], **Antonio Turiel** [1,3] and **Edmo J. D. Campos** [5,6]

1 Department d'Oceanografia Física i Tecnològica, Institut de Ciencias del Mar, Consejo Superior de Investigaciones Científicas, 08003 Barcelona, Spain; olmedo@icm.csic.es (E.O.); pelegri@icm.csic.es (J.L.P.); turiel@icm.csic.es (A.T.)
2 Marine and Environmental Sciences Centre, Faculdade de Ciências da Universidade de Lisboa, 1749-016 Lisboa, Portugal
3 SMOS Barcelona Expert Center, Pg. Marítim 37-49, 08003 Barcelona, Spain
4 Department d'Oceanografia Física i Tecnològica, Institut de Ciencias del Mar, Consejo Superior de Investigaciones Científicas, Unidad Asociada ULPGC-CSIC, 08003 Barcelona, Spain
5 Instituto Oceanográfico, Universidade de São Paulo, São Paulo 01000-000, Brazil; edmo@usp.br
6 American University of Sharjah, P.O. Box 26666, Sharjah, UAE
* Correspondence: castellanos@icm.csic.es or pcossa@fc.ul.pt; Tel.: +34-93-230-95-00

**Abstract:** Three of the world's most energetic regions are in the tropical and South Atlantic: the North Brazil Current Retroflection, the Brazil-Malvinas Confluence, and the Agulhas Current Retroflection. All three regions display offshore diversions of major boundary currents, which define the intensity of the returning limb of the Atlantic meridional overturning circulation. In this work, we use a sea-surface salinity (SSS) satellite product, combined with a high-resolution numerical model and in situ measurements, in order to explore the seasonal variation of the surface currents and transports in these three regions. The analysis of the model output shows that the SSS patterns reflect the surface velocity structure, with the largest horizontal SSS gradients coinciding with those areas of highest velocity and the most predominant velocity vector being 90° anticlockwise (clockwise) from the horizontal SSS gradient in the northern (southern) hemisphere. This information is then applied to the SSS satellite product to obtain maps of water velocity and salt transports, leading to a quantitative tool to estimate both water and salt transports in key regions of the world ocean.

**Keywords:** sea surface salinity; SMOS; retroflections; surface velocity; water transport; salt transport

## 1. Introduction

The long-term variations of the Atlantic meridional overturning circulation (AMOC) can lead to regional changes in the distribution of sea surface temperature (SST) and salinity (SSS) (e.g., [1–3]). Conversely, some key regional-scale processes in the tropical and South Atlantic Oceans can largely influence the dynamics and variability of the returning limb of the AMOC. Of particular relevance to both the regional processes and the overall latitudinal heat and salt transports, are the pathways followed by the returning limb of the AMOC, from the Southern Ocean to the deep-water formation regions in the North Atlantic [4]. While the regions of formation of the North Atlantic Depth Water (NADW) are relatively well known, there are still large uncertainties regarding the origin and pathways followed by the water parcels crossing the South Atlantic into the North Atlantic.

One element that contributes to this high uncertainty is the existence of major regional diversions, commonly named retroflections, where variability is high. These retroflections play an important role on the interconnections among the ocean basins, along the returning AMOC's pathway. In the tropical regions and South Atlantic, there are three key locations where substantial retroflections take place: the Agulhas Retroflection, the Brazil-Malvinas Confluence and the North Brazil Retroflection (Figure 1; top panel).

In the subpolar region, the Malvinas Current (MC) flows towards the equator carrying waters from the Antarctic Circumpolar Current. At approximately 36–38°S, the MC encounters the Brazil Current (BC), forming the Brazil-Malvinas Confluence (BMC), where both subantarctic and subtropical waters collide frontally and are diverted offshore. On the eastern side of the basin, the Agulhas Current (AC) tips along the southern coast of Africa. As the AC surpasses the southern end of the continent, it curls back upon itself before turning East, in a process that leaks rings and filaments into the Atlantic Ocean (Agulhas Leakage, AL) [3,5]. Finally, the North Brazil Current (NBC) also experiences a major retroflection (NBCR) after surpassing the equator along the northeastern coast of South America. This retroflection changes seasonally as waters pile up in the upper interior ocean, and the North Equatorial Counter Current (NECC) eventually connects with the western boundary current [6,7].

Retroflection phenomena are also observed in other regions of the world ocean, usually associated with strong gradients of temperature and salinity. The variability of SSS in these retroflection regions provides useful regional descriptions and, even more importantly, gives insight into the predominant pathways connecting the adjacent ocean gyres. Hence, the continuous monitoring of variables such as SSS is essential to determine and predict the relationship between oceanic and climate variability from regional to global scales. This motivates us to examine the relationship between the SSS and velocity structures in these three western-boundary-retroflection regions. The objective is to examine if we can use the information in the SSS fields not only to describe the variability in regional patterns but also to estimate how the volume, heat, and salt transports change in time, thereby assessing the intensity of the returning limb of the AMOC.

For our analysis, we take advantage of recent spatially and temporally dense SSS measurements obtained using the Soil Moisture Ocean Salinity (SMOS) mission (Figure 1; middle panel), together with outputs from an eddy-resolving simulation with the Hybrid Coordinate Ocean Model (HYCOM) (Figure 1; top panel) and in situ data from the Argo float constellation (Figure 1; bottom panel). We focus on the seasonal variations in the three retroflection regions, investigating the SSS patterns and their coherence with the velocity structures. The paper is structured as follows. In Section 2, we present the data and methods. The variability of the SSS in the retroflection regions is examined in Section 3, the connection between the horizontal SSS-gradient and the velocity fields is developed in Section 3.2. In Section 3.3, we use the above results to calculate the seasonal variability in the water and salt transports associated with the three retroflection regions, and we close the article with some concluding remarks in Section 4.

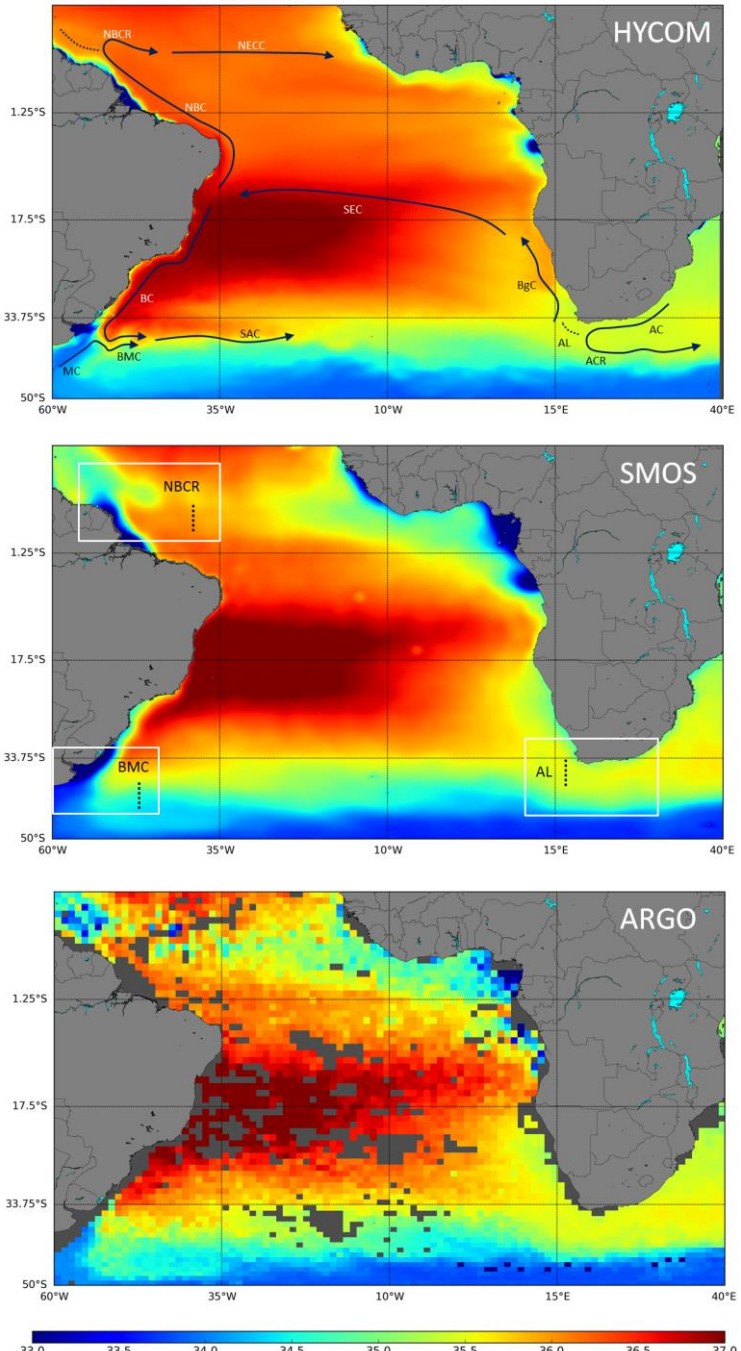

**Figure 1.** Mean sea-surface salinity (SSS) from January 2011 to December 2015 for (**top panel**) Hybrid Coordinate Ocean Model (HYCOM) at 1/12° resolution, (**middle panel**) Soil Moisture Ocean Salinity (SMOS) at 1/4° resolution, and (**bottom panel**) Argo at 1° resolution. The top panel shows a scheme of sea surface currents, highlighting those features relevant to our study: North Brazil Current (NBC), North Brazil Current Retroflection (NBCR), North Equatorial Countercurrent (NECC), South Equatorial Current (SEC), Brazil Current (BC), Malvinas Current (MC), Brazil-Malvinas Confluence (BMC), South Atlantic Current (SAC), Benguela Current (BgC), Agulhas Current (AC), Agulhas Current Retroflection (ACR), and Agulhas Leakage (AL). The middle panel includes rectangles that locate our three study areas, with dotted lines denoting the sections used for the zonal transport calculations.

## 2. Data and Methods

### 2.1. HYCOM Simulation

We have used the output from an eddy-resolving (1/12°) implementation of HYCOM for the Atlantic and Indian Oceans, in a domain extending from 98°W to 114°E and from 65°S to 65°N. The numerical experiment was run at the Ocean Modeling Laboratory (LABMON) of the Oceanographic Institute of the University of Sao Paulo (IOUSP). HYCOM is a primitive-equation hybrid-coordinate ocean general circulation model [8,9].

The model's products analyzed here correspond to the period from 2011 to 2015, extracted from the experiment forced with monthly-mean fields of long-wave radiation, short wave radiation, precipitation and specific humidity, as deduced from NCEP/NCAR reanalysis [10], (Figure 1). The model computes evaporation and sensible heat flux from the precipitation and air temperature data. In the experiment, the SST was relaxed to the climatology, but the SSS was allowed to evolve freely. The bathymetry was extracted from the ETOPO 5 (Data Announcement 88-MGG-02, NOAA, National Geophysical Data Center). During the past decade, HYCOM has become a widely used ocean circulation model, being validated in numerous ocean climate investigations. A complete list of references is available at the HYCOM consortium website, additional information about the experiment and its validation can be found in [11].

### 2.2. SMOS Data

The SMOS mission is an innovative Earth Observation satellite, launched on November 2009, to remotely sense soil moisture over land and sea surface salinity over the ocean [12,13]. The SMOS single payload is the Microwave Imaging Radiometer using Aperture Synthesis (MIRAS), an L-band 2D synthetic aperture radiometer, with multi-angular and full polarization capabilities. This new instrument suggested a technological challenge that required the development of dedicated calibration and image reconstruction algorithms [14].

The SMOS SSS maps are produced from five years (2011–2015) of brightness temperatures measured by SMOS and provided by the European Space Agency (ESA), following the methodology explained in [15] (middle panel of Figure 1). As a result of this novel technique, the SMOS SSS maps are devoid of land-sea contamination, recovering more measurements near the coast. A complete description of the methodology as well as an extensive validation of the product can be found in [14]. Daily SSS maps are computed from objective analysis of the SMOS data in a time window of 9 days at 0.25° resolution (more details in Olmedo et al., 2017 [15]).

### 2.3. Argo Data

We have also used in situ salinity data from close-to-surface acquisitions by Argo floats between January 2011 and December 2015. Argo salinities deeper than 10 m and shallower than 0.5 m are removed, the latter because of the possible presence of air bubbles that increase the error of the conductivity measurements.

Every available Argo surface salinity measurement is compared with the corresponding SMOS/HYCOM SSS monthly values. Figure S1 shows the spatial distribution of the root-mean-square differences or errors (RMSEs) between the SMOS/HYCOM and Argo data; given the different resolutions of the HYCOM and SMOS data, we have uniformized this comparison employing for both cases a 1° grid. Figure S2 presents the temporal evolution of the region-averaged RMSEs between SMOS/HYCOM and Argo for each study area.

The comparison between Argo and SMOS/HYCOM salinity data was satisfactory, with region-averaged RMSEs in the ranges of 0.3–0.4, 0.3–0.6 and 0.2–0.3 for the NBCR, BMC and AL, respectively; these errors, when contrasted with the range of SMOS SSS variability, represent relative variations of 5–7%, 3.5–7.5% and 10.5–15.5% for the NBCR, BMC, and AL, respectively. The largest RMSE occurs for HYCOM in the NBCR and BMC regions, undoubtedly because the model uses

climatological forcing that does not incorporate individual events of high river discharge. In contrast, the SMOS values remain moderate or low except in some particular instances in the NBCR, possibly when the temperature of the Amazon River runoff matches the temperature of the boundary current.

## 3. Results and Discussion

### *3.1. Sea Surface Salinity Variability*

In this section we explore the seasonal variability of the SSS, as provided by the monthly SMOS data, and its relation with the surface velocity fields, as derived from the HYCOM model outputs. For this purpose, we first compute the SSS and surface velocity climatologies. All 2011–2015 SMOS data are used to generate the SSS monthly fields at $0.25°$ resolution. Similarly, the 2011–2015 HYCOM surface velocity fields are used to produce monthly SSS and sea-surface velocities at $1/12°$ resolution, which are then averaged in order to generate the fields at $0.25°$ resolution over the same SMOS grid points.

### 3.1.1. North Brazil Current Retroflection

In the Atlantic Ocean, the transport of heat and salt away from the equatorial and tropical regions takes place largely thanks to its western boundary current, the NBC. However, this transport is largely blocked on a seasonal basis, as the NBCR diverts waters offshore in what becomes the origin of the NECC. To understand this poleward transport of properties, we must assess the seasonal cycle of the NBCR.

The Amazon and the Orinoco Rivers discharge almost 20% of the global freshwater river outflow directly to the surface waters of the western equatorial Atlantic [16]. Particularly, the Amazon flushes its waters close to the equator, which are then transported along-slope by the NBC until at least near 6–8°N. At these latitudes, the waters may either continue along-slope as the NBC or be diverted offshore as the NBCR. Hence, we may use the near-surface freshwater plume in order to track the seasonal changes in SSS and its relation to the NBCR [17].

Figure 2 shows the monthly variability of SSS in the NBC and NBCR. The seasonal pattern is clear in the SSS from both SMOS and HYCOM, with the plume of low salinity waters stretching offshore between July and December, reaching its maximum expression in September. These changes in the eastward extension of the low-salinity plume are not directly related to the temporal variation in Amazon River discharge—minimum in October-November and maximum, about 3.5 times greater, in May–June [18]—but rather to the seasonal appearance of the interior NECC [6,7]. For our purposes, the low-salinity plume behaves only as a tracer of the subjacent dynamics. The velocity fields from HYCOM also display the high seasonality of the NBC and NBCR: between January and June, most of the flow continues along-slope while from July to December a large fraction of the flow retroflects offshore between about 5 and 10°N.

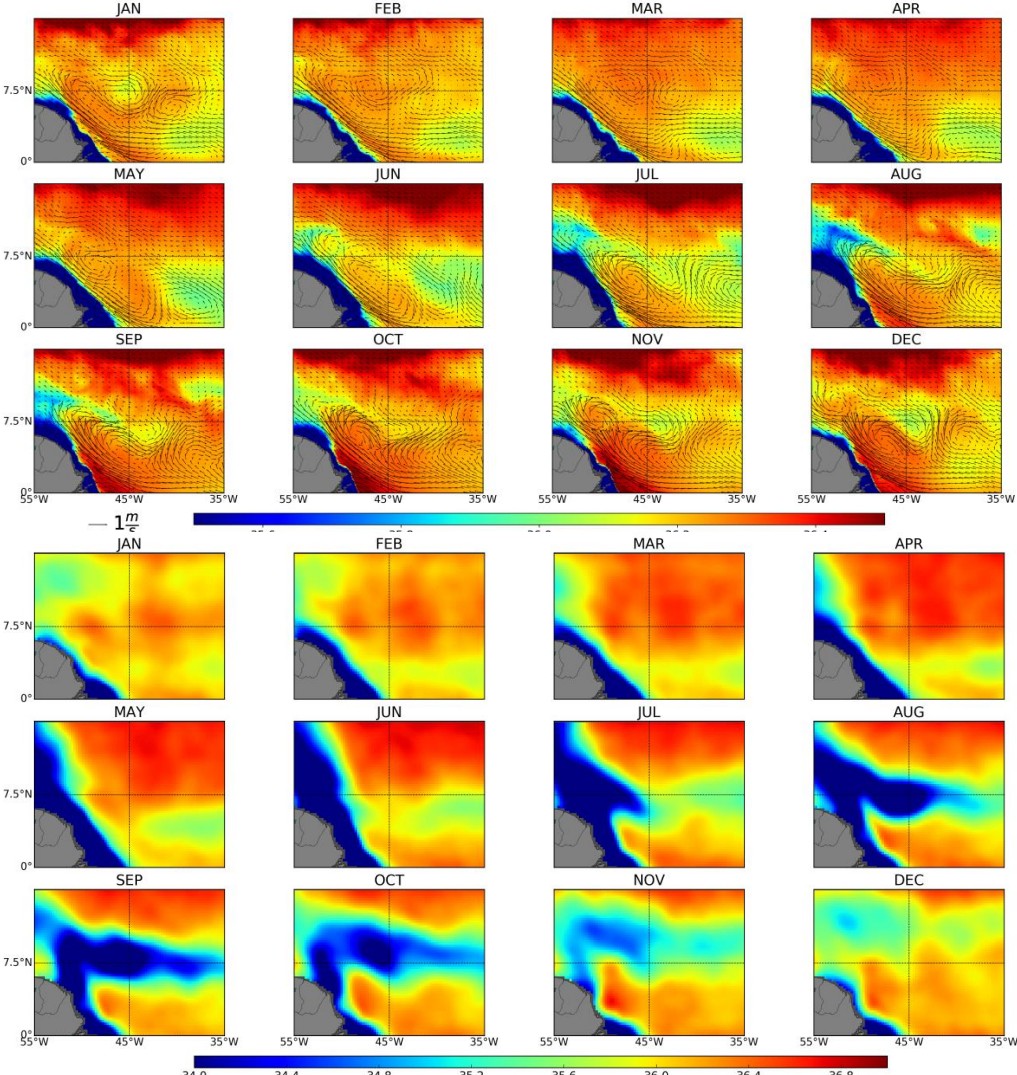

**Figure 2.** (**Top panels**) Monthly mean SSS (colored) and surface velocity field (velocity vectors) from January 2011 to December 2015 for HYCOM at $1/12°$ resolution in the NBCR. (**Bottom panels**) The same using SMOS data but without velocity vectors.

### 3.1.2. Brazil-Malvinas Confluence

The convergence of the BC and MC leads to the BMC, an intense frontal system between warm-salty subtropical and cold-fresh subantarctic waters [19,20]. The subtropical and subantarctic waters are diverted eastward along the frontal system, in a process that sheds numerous eddies of both signs [21]. This may be viewed as a process that enhances eddy-like latitudinal diffusion of both heat and salt.

Despite the averaging over five years, the HYCOM monthly distributions of SSS and velocity illustrate how the encounter between the BC and the MC gives rise to the sharp turn of the less-salty subantarctic waters (the MC retroflection, MCR) and the southward penetration of the salty subtropical waters (the BC overshoot, BCO) [22] (Figure 3, upper panels). These regional patterns also appear, although less clear, in the SMOS SSS fields, certainly because of their more limited spatial resolution.

The SMOS salinity fields show the outflow of La Plata River stretching along the continental margin all year long, although the low-salinity values extend further south between April and August (Figure 3, lower panels); this agrees with the moderate seasonal changes in river discharge (variations of about 50% from the mean, with minimum values in September and December–January, and maximum values in April–May) [23].

According to both HYCOM and SMOS, the frontal system displays moderate variations in latitudinal position. These changes are small near the shelf break but become more visible offshore (east of 52.5°W), with the high subtropical SSS reaching further south during the austral summer (November through March) in both outputs. Despite the different resolutions, both HYCOM and SMOS display similar seasonal patterns of intensification of the BCO (high SSS between about 40 and 44°S), being enhanced in January–April and September–December.

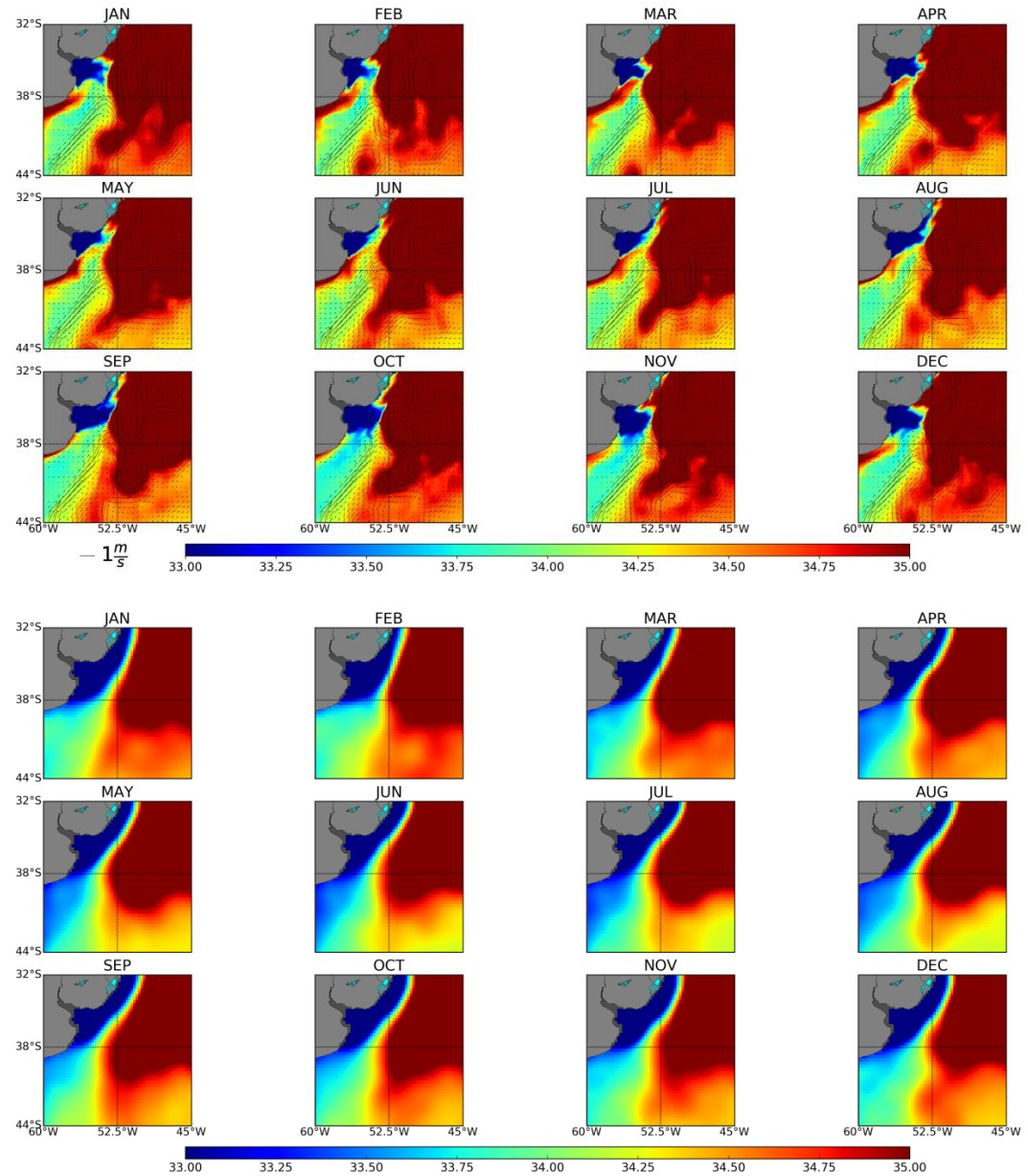

**Figure 3.** As in Figure 2 but for the BMC.

### 3.1.3. Agulhas Leakage

Possibly the most energetic among all open-ocean diversions is the abrupt turning of the Agulhas Current (AC) as this western boundary current enters the Atlantic Ocean, in what has been named the AC retroflection (ACR) [5]. The counterpart of the ACR is the AL, either as rings or filaments, whereby some Indian Ocean waters become incorporated into the South Atlantic. The stronger the AL is, the larger transports of salt and heat between the Indian and Atlantic Oceans, as part of the returning AMOC limb [24].

Maps of SSS and surface velocity provide information on the regional dynamics in the AL region (Figure 4). Despite the ubiquity of the ACR, with some mesoscale variability probably related to the relatively small number of years used for calculating the monthly averages, both the SMOS and the model display a seasonal cycle in SSS for a zonal band southwest of Africa—from the continent to about 40°S and stretching between 10°E and 20°E—where the AL is to occur. The SMOS data show maximum SSS values between December and April, and minimum values between June and September. However, the seasonal appearance of high SSS values is more related to the oscillation in the longitude of retroflection than to the intensity of the AL [25–27].

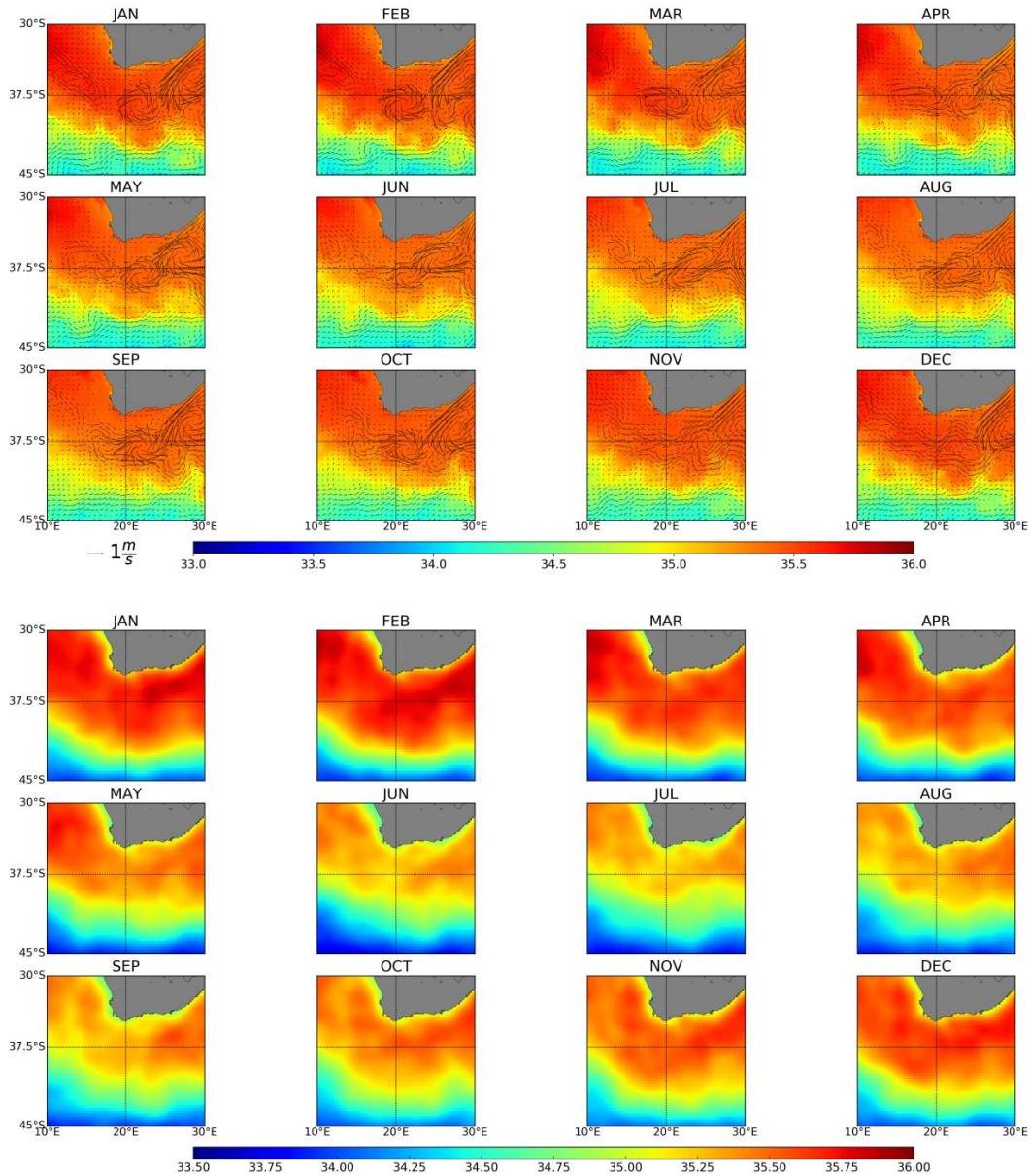

**Figure 4.** As in Figure 2 but for the AL.

### 3.2. Relation Between the Surface Salinity and Velocity Fields

The salinity fields are related to the velocity fields in two different ways. First, if the water parcels conserve salinity, then the contours of constant salinity will coincide with trajectories; if the field is steady, these constant-salinity contours will also coincide with the streamlines. Second, if we consider the deep ocean to be in geostrophic balance then the velocity field will be normal to the pressure

gradients; in a baroclinic ocean this pressure gradient is related to both the temperature and salinity gradients. Both considerations point at the relevance of calculating the horizontal SSS gradient and comparing it with the velocity field.

Regarding the above second point, it is worth emphasizing that both salinity and temperature increase towards the sea surface for most of the Atlantic Ocean. Consequently, both the SSS and SST gradient vectors will usually be anti-parallel to the pressure-gradient force, so that, for sea-surface geostrophic velocities, the angle between the SSS gradient vector and the velocity vector will be 90° (measured from the former to the latter) in the northern hemisphere and 270° in the southern hemisphere. Another related issue is that temperature and salinity variations oppose each other in setting the density changes; in the upper ocean, temperature has always a greater effect than salinity in setting the density but the relation between both may change from one region to another. Hence, the SSS gradient vector is indeed an indicator of the intensity and direction of the velocity field, although the precise relation will change from place to place, to the point that in some areas the speed may be directly proportional to the amplitude of the salinity gradient, while in other places there may be an inverse relation.

In this section, we use the HYCOM model salinity and velocity outputs to infer, for each of the three retroflection regions, the functional relationships that best relate both fields. These dependences will then be applied to the SSS SMOS data in order to infer the surface velocities and transports (Section 3.3). The procedure, done separately for each region and month, has three steps. First, we produce the monthly climatologies of surface horizontal velocity and SSS, and, from the latter, we compute the SSS horizontal gradients using center differences (Figures S3–S5, Supplementary Materials; hereafter we will always refer to the horizontal components of the velocity and the SSS gradient). Secondly, we produce a probability density function of surface water speed $V$ for each absolute value of the SSS gradient $|\nabla S|$. The functional relation $V = f(|\nabla S|)$ is then set as the maximum probability value of the speed $V$ for each value of $|\nabla S|$ (Figures 5–7, top panels). Finally, we compute the angle between the SSS-gradient and velocity vectors (measured anticlockwise from $\nabla S$) and produce a frequency distribution of the occurrence of each angle; the angle that sets the relationship is then selected to be the most frequent one, $\theta_{max}$ (Figures 5–7, bottom panels), and the angle between $\nabla S$ and the eastward direction is given by $\theta = \theta_{gs} + \theta_{max}$, where $\theta_{gs}$ correspond to the orientation of $\nabla S$.

Hence, the zonal and meridional components of the velocity field, may be written as follows:

$$u = f(|\nabla S|) \cos\theta \tag{1}$$

$$v = f(|\nabla S|) \sin\theta \tag{2}$$

In the NBCR, the most frequent argumental difference between the SSS gradient and velocity vectors is 90° (Figure 5); in this region, the functional relation between the speed and the amplitude of the salinity gradient is not lineal. Regarding the BMC, the argument histogram is bi-modal, though the most probable value is 270° (Figure 6), so we set this value as the constant difference between the SSS gradient and velocity vectors. The existence of a second peak at about 90° is caused by the presence of relatively fresh Subantarctic Shelf Waters along the Patagonian shelf [28], which causes the SSS gradient vector to be locally parallel to the pressure gradient force (Figure S6, Supplementary Materials). For both the BMC and NBCR regions, the relationships between speed and the amplitude of the salinity gradient are quite lineal.

In the AL region (Figure 7), we have also considered the angle between the SSS gradient and velocity vectors as constant and equal to 270°. An interesting situation happens, we observe a decreasing relation between the SSS gradient and the velocity (see Figure 7, top). This decreasing relation happens because the SSS gradient is compensated by the temperature gradient. The compensation between the temperature and salinity gradient typically occurs in the regions where the horizontal mixing dominates the dynamics [29]. The AL region is a region where eddies

are generated, which it is increases the eddy diffusivity. Therefore, we could say that we are in the previous situation.

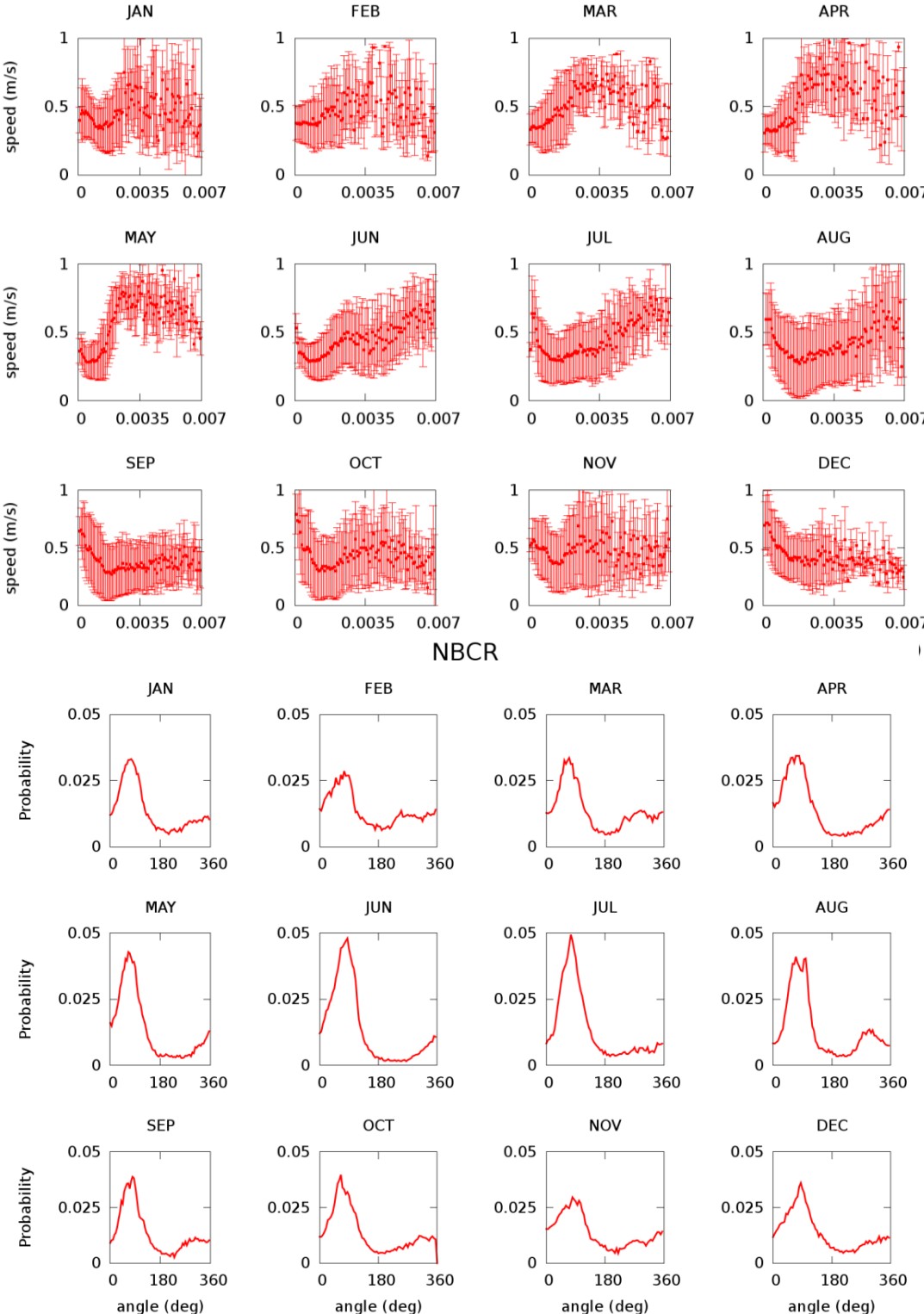

**Figure 5.** Monthly dependence of the horizontal speed of water and the SSS horizontal gradient in the NBCR, as inferred from HYCOM. (**Top panels**) Maximum probability value and standard deviations for water speed as a function of the absolute value of the SSS gradient. (**Bottom panels**) Frequency distribution of the angle between SSS gradient and surface water velocity vectors (measured from the former to the latter).

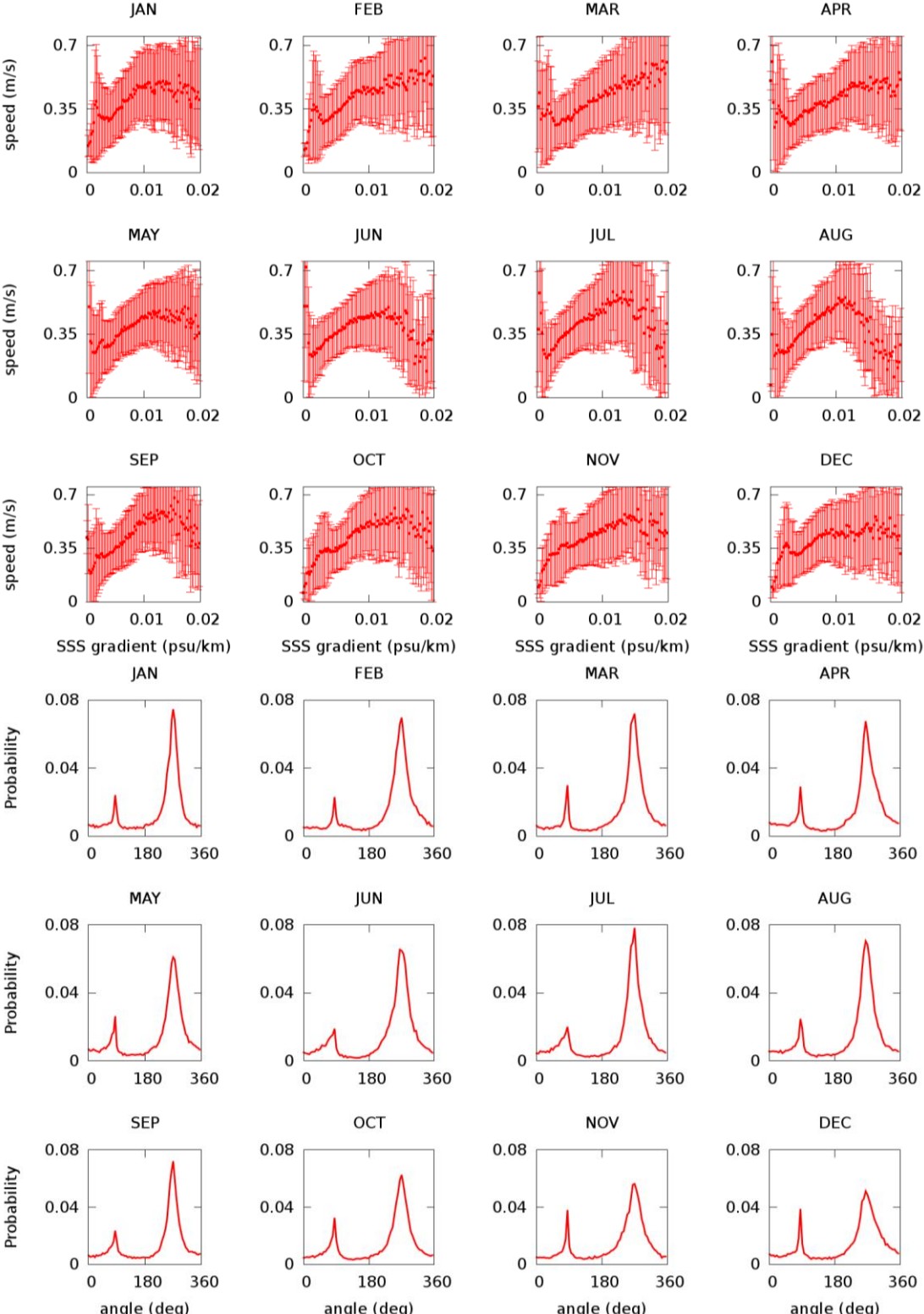

**Figure 6.** As in Figure 5, but for the BMC.

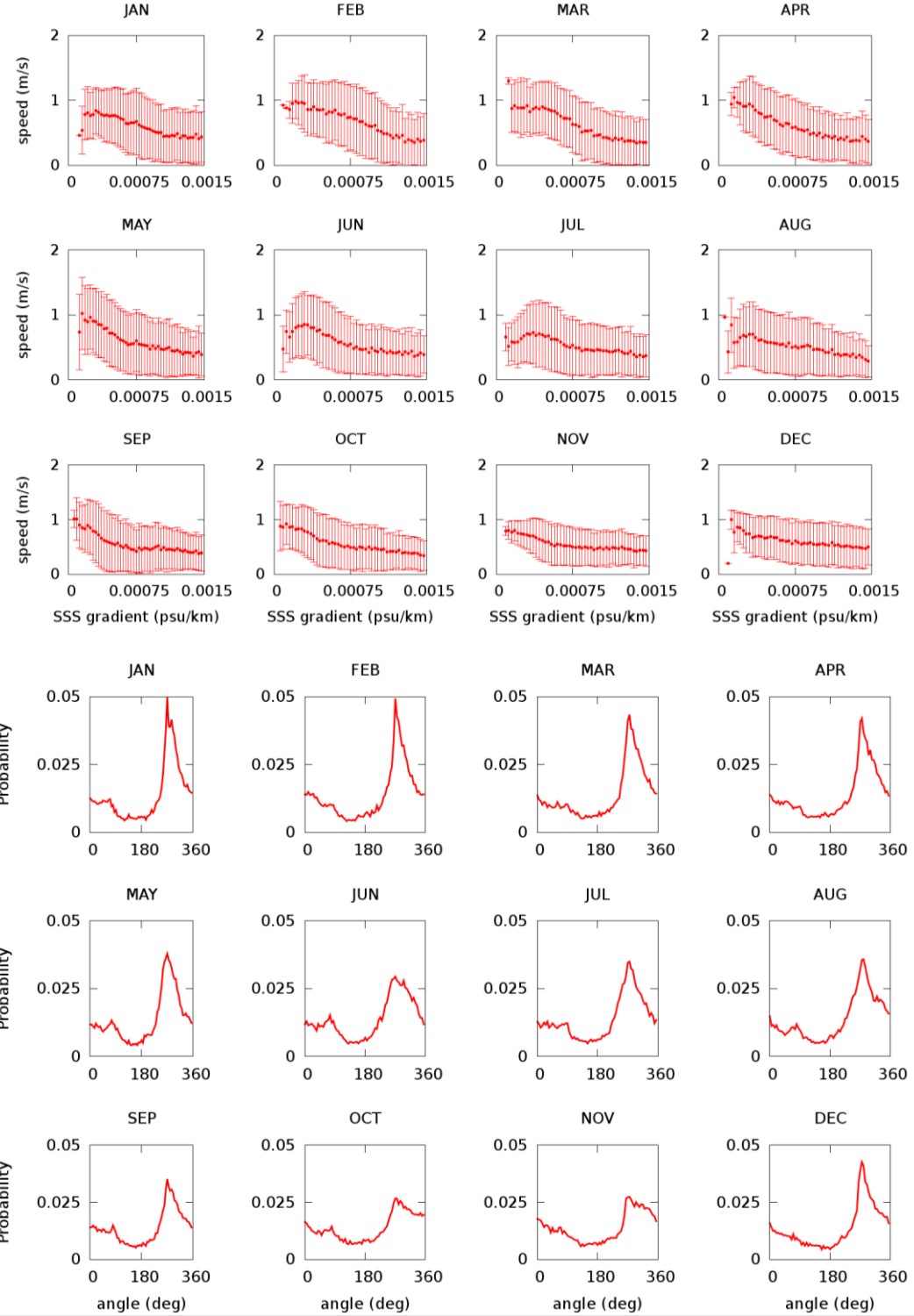

**Figure 7.** As in Figure 5, but for the AL.

Finally, the procedure presents some limitations that are worth mentioning. One drawback is that the amplitude of the SSS gradients is smaller in the model than as deduced from SMOS, which will likely lead to an underestimation of the velocities. Another limitation is that we may sometimes find very different adjacent regimes, thereby constraining the application of the method to one selected region. One such example is the southern edge of the ACR, which is characterized by very large salinity gradients because of the influence of the circumpolar current; hence, in this work we will

restrict the method to the northern and core regions of the ACR. Nevertheless, the results are very encouraging, both in terms of the direct relation between speed and the amplitude of the SSS gradients as well as the agreement of the most frequent angles (either 90° or 270°) with what we would expect for geostrophic currents.

### 3.3. Salt Transport Variability

In last section we developed a simple methodology to infer the surface velocities from the SSS fields, which we now apply in order to obtain the seasonal cycles of water and salt surface transports associated with the three retroflections. This calculation is entirely done using the SMOS SSS fields, both to infer the velocities and integrated water transports and also to obtain the salt fluxes and integrated salt transports. The salt fluxes are simply calculated as the product of velocity, SSS and an average surface density (here taken as 1025 kg m$^{-3}$).

The velocity and salt fluxes characterize the surface layers so that the integration is done in one horizontal direction, thereby obtaining transports per unit depth over selected sections. The changes in SSS are much smaller than the changes in surface velocity so the monthly patterns of velocity and salt flux (not shown) are very much alike. Similarly, the seasonal variations of the water and salt integrated transports follow analogous variations (Figure 8). We must keep in mind that the subsurface velocity and salinity fields often bear similarities with the surface ones. Hence, our results may likely reflect the character of the depth-integrated transports—i.e., a water transport of $10^5$ m$^2$ s$^{-1}$ over a depth of 100 m would represent a water transport of 10 Sv—but this calculation falls beyond the objectives of our study.

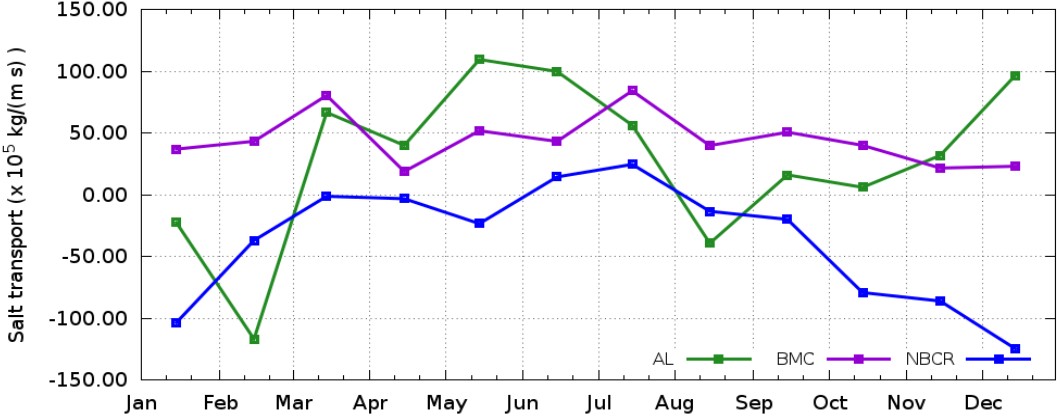

**Figure 8.** Salt zonal transport per unit depth for the North Brazil Current Retroflection (NBCR), the Brazil-Malvinas Confluence (BMC), and the Agulhas Leakage (AL). In all cases, the integration is over 4° of latitude, as explained in the text, with positive/negative values denoting eastward/westward transports.

### 3.3.1. North Brazil Current Retroflection

When considering the NBC, we are interested in assessing how much water and salt are diverted zonally by the NBCR, towards the interior tropical North Atlantic Ocean (Figure 1). The stronger this diversion, the weaker will be the meridional transfer of heat and salt from the tropical Atlantic Ocean towards the temperate North Atlantic waters.

In order to assess the monthly strength of the NBCR, we integrate the zonal velocity and salt flux along 40°W between 2°N and 6°N (Figure 1). The resultant water and salt zonal transports show a clear seasonal pattern, with intense westward transports between October and January that revert in June and July (Figure 8). This pattern is characteristic of the year-long dominance of the westward SEC except during the summer months, when the NBCR develops and connects to the downstream eastward NECC [6,7,30].

### 3.3.2. Brazil-Malvinas Confluence

The encountering of the BC and MC sets up the BMC, a remarkable frontal system that behaves as a zonal guide from the western boundary to the interior South Atlantic Ocean. The flow along this front represents the origin of the South Atlantic Current (SAC), carrying both warm-salty subtropical and fresh-cold subantarctic waters (Figure 1). This current is characterized by the presence of numerous eddies of both signs, with subtropical (warm and salty) eddies moving southwards and subantarctic (cold and fresh) eddies following north. The more intense the BMC and SAC, the larger the eddy activity will be, thereby increasing the meridional heat exchange.

In order to evaluate the monthly intensity of the flow emerging zonally at the BMC, we integrate the zonal velocity and salt flux along 47°W between 40°S and 44°S (Figure 1). The latitudinally-integrated water and salt zonal transports show fairly constant eastward values for all months. These values are indicative of an intense BMC throughout the year, with latitudinal BC and MC transports that do not display a prominent seasonal cycle (Figure 8) [31]. These currents converge onto the frontal system and are diverted eastwards to set up the origin of the SAC [32–35].

### 3.3.3. Agulhas Leakage

The returning limb of the AMOC depends largely on the ACR and AL, as the relative strength of both currents controls the transfer of warm and salty Indian Ocean waters into the South Atlantic Ocean. The AC cannot overtake the southern tip of Africa and returns east as the ACR. The actual penetration proceeds intermittently as rings and filaments, in what is known as the AL, connecting the AC with the equatorward Benguela Current (BgC).

In order to assess the seasonal variation of the Agulhas Leakage (AL), we integrate the zonal velocity and salt flux along 17°E between 34°S and 38°S (Figure 1). The subsequent water and salt zonal transports display substantial intermittency (Figure 8). This agrees fairly well with observations that show no prominent seasonal pattern in the AL, with the actual monthly transports changing as a function of the shedding of rings from the AC [36,37].

## 4. Conclusions

For conditions of weak vertical and horizontal mixing, and in the absence of intense evaporation or precipitation, surface water parcels approximately conserve their salinity and SSS serves as a good indicator of the surface flow field. This result is the case for relatively short time scales, shorter than the time scale that characterizes both the local rate of change and advection through the area of study. In these circumstances, the flow is nearly stationary and the constant salinity contours coincide with both trajectories and streamlines. Hence, the sources of high (e.g., Indian Ocean) or low salinity (e.g., Amazon River discharge) serve to delineate the flow field, with the velocity vector tangent to the constant SSS contours.

There is still a second justification for using the SSS as an indicator of the flow field. The subinertial open-ocean flow (with time scales longer than one day) is largely in geostrophic balance: the velocity field is normal to the contours of constant pressure, which often closely parallel the temperature and salinity contours. Further, the greater the pressure gradient, i.e., the closer the constant pressure contours are, the faster the velocity fields.

Both above reasons point at the possibility of inferring the velocity field from the constant-salinity contours, as long as we do our analysis on the proper time scale: longer than inertial and shorter than the advective and local rate-of-change time scales. The local rate-of-change time scale is often dominated by the seasonal cycles, so a reasonable time scale is one month. Using one month as the advective time scale, and considering a swift ocean flowing at 0.2–0.4 m s$^{-1}$, leads to considering regions no larger than 1000–2000 km approximately.

We have used these ideas to explore if we can use the SSS fields to characterize the flow in three retroflection regions with typical lengths of 1000–2000 km. Our approach has consisted, first, in using

a well-contrasted ocean model to infer monthly functional relations between SSS and surface velocity and, second, in applying these relationships to the SMOS data, an independent and novel SSS data set. Our results are encouraging, as we have obtained rather robust functional dependences between the model SSS and surface velocity, and because the most-frequent angle between both vectors is consistent with what we expect for geostrophic currents. These relations, when translated to the SMOS data, lead to consistent patterns of seasonal variability for the water and salt transports associated with the retroflections.

Future research should involve using long model outputs for analyzing the relationships between SSS gradients and surface velocities at different temporal scales, from shorter time scales in more local sites to longer time scales over much larger areas. This will allow a better understanding of the dynamics behind each functional relation, thereby providing a better assessment of the advantages and limitations of this approach. Once the relationships are well established, and longer SMOS time series become available, we will have the capacity to identify the inter-annual changes in water and salt transport for different ocean domains, including the critical western-boundary retroflection regions.

**Supplementary Materials:** The following are available online at http://www.mdpi.com/2072-4292/11/7/802/s1, Figure S1. (Left panels) Root Mean Square Error (RMSE) in SSS between HYCOM and Argo in a $1° \times 1°$ grid, in (top) the North Brazil Current Retroflection, (middle) the Brazil-Malvinas Confluence and (bottom) the Agulhas Leakage. (Right panels) As in the left panels but using SMOS and Argo data. Figure S2: Time evolution of the RMSE in SSS between HYCOM and Argo data (dashed lines) and between SMOS and Argo data (solid lines), for (top panel) the North Brazil Current Retroflection, (middle panel) the Brazil-Malvinas Confluence and (bottom panel) the Agulhas Leakage. Figure S3: Monthly climatology in the North Brazil Current Retroflection from HYCOM data. (Top panels) Speed of the surface water. (Bottom panels) Absolute value of the SSS horizontal gradient. Figure S4: Monthly climatology in the Brazil-Malvinas Confluence from HYCOM data. (Top panels) Speed of the surface water. (Bottom panels) Absolute value of the SSS horizontal gradient. Figure S5: Monthly climatology in the Agulhas Leakage from HYCOM data. (Top panels) Speed of the surface water. (Bottom panels) Absolute value of the SSS horizontal gradient.

**Author Contributions:** Conceptualization, P.C., E.O., J.L.P. and A.T.; Investigation, P.C. and E.O.; Methodology, P.C., E.O., A.T. and E.J.D.C.; Writing – original draft, P.C. and E.O.; Writing – review & editing, J.L.P. and E.J.D.C.

**Funding:** This work has been funded by the Spanish government through the National R+D Plan through projects VA-DE-RETRO (reference number CTM2014-56987-P), Promises (reference number ESP2015-67549-C3) and L-Band (reference number ESP2017-89463-C3-1-R). The HYCOM numerical experiment was supported by the São Paulo State Foundation for Research Support (FAPESP, Grants: 2008/58101-9, 2010/01943-8, and 2011/50552-4). E. Campos acknowledges the Brazilian National Council for Scientific and Technological Development (CNPq) for a Research Fellowship (Grant 302018/2014-0) and FAPESP (Grants, 2017/09659-6).

**Acknowledgments:** The Argo data were collected and made freely available by the International Argo Program that contributes to it (http://www.argo.ucsd.edu, http://argo.jcommops.org). The Argo Program is part of the Global Ocean Observing System. Finally, we are very grateful to our two reviewers for a number of useful comments and suggestions.

**Conflicts of Interest:** The authors declare no conflict of interest.

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
