# Peer review of "Seasonal Variability of Retroflection Structures and Transports in the Atlantic Ocean as Inferred from Satellite-Derived Salinity Maps"

_remotesensing, doi:10.3390/rs11070802_

Round 1

Reviewer 1 Report

A very interesting topic is being presented in this paper.

The writers need to make the flow of ideas a bit smoother than it already is.

They were able to put years of research and effort spend in producing reliable salinity product from satellite microwave equipment to use.

I attached my comments.

I need to read this one more time before submission.

Best Regards, 

Author Response

We would like to sincerely thank our reviewer for his/her useful and constructive comments.The point by point response in the attach.

Thank very much for your time,

Paola Castellanos

Reviewer 2 Report

This paper describes calculations of transport and seasonal variability of SSS in specific regions of the tropical and South Atlantic. The regions studied, the Brazil-Malvinas confluence, the North Brazil Current Retroflection and the Agulhas Leakage, are important points for understanding the dynamics of the basin. The paper presents a novel approach to determining the velocity field from SSS and fluxes across specific lines (Fig. 1b) from it. The paper is well-crafted and easy to read and thus I strongly recommend publication. I have made some specific suggestions below that may improve it.

Several points below require input from the editor concerning RS's policies.

Nowhere in the paper does it say where the data could be obtained from. A URL or DOI are needed for both the Argo and SMOS datasets, as well as the HYCOM output the authors used. The generic links to the argo website (line 371) or HYCOM (line 93), or references to methodology descriptions (line 406) are not sufficient. The editor can clarify as to RS's policy.

Line 59. "velocity structures"

Line 84. Probably "Data and Methods" would be a better title.

Lines 116-117. This belongs in the acknowledgements.

Line 117. www.argo.ucsd.edu

Lines 120-121. Did the authors do the averaging themselves, or did they use one of the many Argo interpolated products available? Either way, more details are needed about how these data were treated.

Line 122. 0.4. Are these RMSEs for HYCOM, SMOS or both?

Section 3.1. Shouldn't this material be in section 2?

Line 147. A reference to Grodsky et al would be appropriate here.

JOURNAL OF GEOPHYSICAL RESEARCH: OCEANS, VOL. 119, 1–12, doi:10.1002/2013JC009450, 2014

Line 167. "greater"

Line 196. "mesoscale"

Line 219. I do not understand why these figures are being called "Supplementary materials". They are referred to and discussed in the text. Perhaps the editor can clarify what RS's policy is with regards to SM.

Line 227. "towards" what?

Line 230. These are not the zonal and meridional components. They appear to be the across and along-gradient components given that theta-max is measured relative to grad-S (line 227).

Lines 224-230. This definition is unclear. u and v are components of velocity at each grid point? Or perhaps they are functions at each point? Setting a unitful quantity, V, equal to a unitless function, the pdf, on line 225 is confusing. An example or a schematic picture would help.

Figures 5-7. This must be some kind of regional average. It should be specified.

Lines 231-236. Cool! This is a fascinating way to look at the relationship between velcoity and SSS gradient. Several issues come out that are worth mentioning. The fact that the peaks are at 90deg for the NBCR (northern hemisphere) and 270deg for the other two regions (southern hemisphere) is what we would expect. (I wish the angular scale would go from -180 to 180 instead of 0 to 360). What is interesting is that in the BMC there are also peaks at 90deg. These would seem to be anticyclonic eddies with low SSS in the core (or vice versa). Would the authors please discuss these secondary peaks and tell us where and when they might occur, and how they relate to the BMC. The widths of the peaks give some indication of the departure from geostrophy. This is larger for the NBCR than the other two regions. For the AL, the peak is very seasonal, and almost disappears in late austral winter. Perhaps the authors can discuss the seasonality of the peaks and the departure from geostrophy. Some of this discussion goes on in the conclusions section (lines 336-344), but it would be nice to see this more fully fleshed out.

Lines 236-237. This is a very surprising and counterintuitive result. One would have expected the surface velocity to increase with increasing SSS gradient. The SSS is at least partially responsible for the surface density, which should control the surface pressure, and thus the velocity field. Can the authors please explain why this would be. See also lines 328-329.

Figures 8a and b are so much alike it hardly seems necessary to include both. What do the positive and negative values mean?

Line 318. Or vertical or horizontal mixing.

Line 323. The Amazon River.

Line 333. The local rate of change would likely be dominated by the mesoscale and is much shorter than this.

Lines 353-364. This must be here to satisfy the editorial requirements of RS, but it is repetitive and superfluous.

Author Response

We would like to sincerely thank our reviewer for his/her useful and constructive comments.
We appreciate your corrections and we have made an effort to response your suggestions.

Thanks very much for your time,

Paola Castellanos

Round 2

Reviewer 1 Report

The writer had response to all my questions clearly, and have  done the requested modifications.

In my humble opinion, I think this paper is good for publishing.

Author Response

We are grateful for your contribution.

Paola Castellanos

Reviewer 2 Report

This is a second review of this paper, mainly focusing on my previous comments and whether the authors have adequately responded to them. There are a number of areas, where, in my opinion, they have not. Details below.

The title of section 2 was not changed.

Section 2.3 second paragraph. The authors have apparently compared individual argo surface measurements with monthly values from SMOS and HYCOM. This is not a valid comparison because of the different amounts of averaging and the time scales involved. The data are then compared on a 1X1 grid instead of the 0.25deg grid for SMOS or the 1/12deg grid of HYCOM.

Fig. 2 caption. "(Top panels) Monthly mean SSS from January 2011 to December 2015 for HYCOM at 1/12deg resolution in the NBCR. Overlaid velocity vectors are also from HYCOM. (Bottom panels) The same using SMOS data, and with no overlaid velocity field."

Fig. 2 Are the color scales in each set of panels the same? If so, only one is necessary. The numerical values for the top panel are illegible. The HYCOM velocity vectors need to include a scale arrow somewhere.

Figs. S3-5. These need units.

"Secondly, we produce a probability density function of surface water speed V..." I guess these are averaged over the regions shown in Fig. 1(middle), but should not have to guess.

"Hence, the zonal and meridional components of the velocity..." As I said in my previous review, these are not the zonal and meridional components. I'm not sure why the authors have refused to fix this.

I guess "max" is a subscript of theta in equations (1) and (2).

"Also a a secondary peaks..." Fix the grammar in this sentence.

The authors have not really done much to explain these secondary peaks, despite my request to do so. They are interesting, and counterintuitive and deserve closer scrutiny. There must be an unusually large countervailing SST gradient that offsets the SSS gradient in the eddies where this occurs.

The authors have not explained the decreasing relationship between speed and gradient in Fig. 7, despite my request to do so.

The authors have left in Fig. 8b in spite of my recommendation to delete it, or Fig. 8a. There is no need for both.

Author Response

Thank you very much for your comments.  We appreciate your contribution.The answers in attach.

Paola Castellanos

Round 3

Reviewer 2 Report

This is my third review of this paper. The authors have responded positively to many of the requested changes from my previous review and thus it is mostly ready for publication. However, they have not really addressed a few issues that are detailed below.

Fig. 2 caption. Remove "SSS from SMOS data". Redundant.

Fig. 2. The two sets of panels have a very different color scale. This makes it look like HYCOM has a much stronger low SSS signal of the NBCR. I would recommend matching color scales to make comparison easy. (In Fig. 3 the color scales match, while in Fig. 4 they almost do.)

Section 2. The paper does not say anywhere where the data can be obtained. I do not know what the editorial policy of RS is, but in many journals, a URL or DOI would be required. Also, the authors need to state exactly what version of the SMOS data they use, and what version of HYCOM.

Section 3.1.2. "The more unstable of this..." I'm not sure what this is trying to say.

Section 3.1.3. "Notice, however, that the seasonal appearance..." What am I supposed to notice?

Section 3.2. "Hence, the SSS gradient vector is indeed an indicator..." I guess this is in response to my request for an explanation of why SSS gradient may have a decreasing influence on velocity in the Auglhas leakage region (Fig. 7). However it is not very satisfactory. The authors are essentially saying that it's that way because we observe it to be so, a circular argument. What is it about T and S gradients that is different here than the other two regions? Why would sharper SSS gradients be associated with slower velocity? Perhaps a schematic picture or an example would help. It's a curious and counterintuitive result and the reader will want to know how to interpret it.

Section 3.2. "...zonal and meridional components..." Again, these are not the zonal and meridional components, they are the across and along-gradient components. I understand that theta is measured relative to the gradient vector, not the north-south-east-west direction. If this is not a correct understanding, then the authors need to rewrite the sentence starting with "Finally, we compute the angle..." as it is stated there.

Fig. S6. This figure shows nicely how the extra peak comes about, being confined to the Patagonian shelf. It's a great addition to the paper. However, there is a little more detail that needs to be included. Is this an average picture or a snapshot? Either way it should be stated and more details given. E.g. what date is the snapshot for? Is the domain depicted the same as Fig. 3? No lat/lon coordinates are given for this picture.

Section 3.3. The authors do not say how velocity is inferred from SMOS SSS. Given the discussion just above in this review, this may be problematic, especially in the AL region.

Reference 17. "curious"

Author Response

Thanks very much for your contribution.

Fig. 2 caption. Remove "SSS from SMOS data". Redundant.
Corrected, thanks

Fig. 2. The two sets of panels have a very different color scale. This makes it look like HYCOM has a much stronger low SSS signal of the NBCR. I would recommend matching color scales to make comparison easy. (In Fig. 3 the color scales match, while in Fig. 4 they almost do.)

Section 2. The paper does not say anywhere where the data can be obtained. I do not know what the editorial policy of RS is, but in many journals, a URL or DOI would be required. Also, the authors need to state exactly what version of the SMOS data they use, and what version of HYCOM.

Thanks for your comments, the version of SMOS product is indicated in the last version of manuscrit. The HYCOM simulation is from LABMON, Edmo Campos runned the simulation, more details in Castellanos et al., 2015.

Section 3.1.2. "The more unstable of this..." I'm not sure what this is trying to say.

Corrected, thanks

Section 3.1.3. "Notice, however, that the seasonal appearance..." What am I supposed to notice?

Corrected, thanks

Section 3.2. "Hence, the SSS gradient vector is indeed an indicator..." I guess this is in response to my request for an explanation of why SSS gradient may have a decreasing influence on velocity in the Auglhas leakage region (Fig. 7). However it is not very satisfactory. The authors are essentially saying that it's that way because we observe it to be so, a circular argument. What is it about T and S gradients that is different here than the other two regions? Why would sharper SSS gradients be associated with slower velocity? Perhaps a schematic picture or an example would help. It's a curious and counterintuitive result and the reader will want to know how to interpret it.

We have included a discussion in this new version of the manuscript (pages 11). Thanks.

Section 3.2. "...zonal and meridional components..." Again, these are not the zonal and meridional components, they are the across and along-gradient components. I understand that theta is measured relative to the gradient vector, not the north-south-east-west direction. If this is not a correct understanding, then the authors need to rewrite the sentence starting with "Finally, we compute the angle..." as it is stated there.

Thank you very much. The reviewer is completely right. There is a    typo in the formulation \theta=\theta_{\grad SSS}+\theta_{max}
 Therefore, the final reference of the angle is  north-south-east-west... so we are actually computing zonal and  meridional components. We have corrected the formulation in this new version of the manuscript.

Fig. S6. This figure shows nicely how the extra peak comes about, being confined to the Patagonian shelf. It's a great addition to the paper. However, there is a little more detail that needs to be included. Is this an average picture or a snapshot? Either way it should be stated and more details given. E.g. what date is the snapshot for? Is the domain depicted the same as Fig. 3? No lat/lon coordinates are given for this picture.

Details have been included in the legend of the figure, thanks.

Section 3.3. The authors do not say how velocity is inferred from SMOS SSS. Given the discussion just above in this review, this may be problematic, especially in the AL region.

The discussion of this section in the new version of manuscript  respond of this, thanks.

Reference 17. "curious"

Corrected, thanks

This manuscript is a resubmission of an earlier submission. The following is a list of the peer review reports and author responses from that submission.